# Characterization of a Novel Infectious Pancreatic Necrosis Virus (IPNV) from Genogroup 6 Identified in Sea Trout (*Salmo trutta*) from Lake Vänern, Sweden

**DOI:** 10.3390/vetsci10010058

**Published:** 2023-01-14

**Authors:** B. David Persson, Jacob Günther Schmidt, Mikhayil Hakhverdyan, Mikael Leijon, Niels Jørgen Olesen, Charlotte Axén

**Affiliations:** 1National Veterinary Institute, 751 89 Uppsala, Sweden; 2National Institute of Aquatic Resources, Technical University of Denmark, 2800 Kongens Lyngby, Denmark

**Keywords:** infectious pancreatic necrosis virus, IPNV genogroup 6, infection trial, salmon, sea trout, rainbow trout

## Abstract

**Simple Summary:**

Infectious pancreatic necrosis virus (IPNV) causes serious disease in salmonid species, especially in fry. In 2016, we identified an IPNV infected sea trout in Lake Vänern in Sweden. Lake Vänern hosts four small populations of landlocked Atlantic salmon and sea trout that each migrate annually to two different rivers to spawn. The virus was isolated and after sequencing placed it in genogroup 6. The pathogenicity of the virus was investigated in an infection trial in Atlantic salmon, sea trout, and rainbow trout. In the trial, the novel isolate was compared to an isolate of genogroup 5, that usually cause high mortalities. By the end of the trial, the Swedish genogroup 6 isolate only caused mild infection in all three species, suggesting a low pathogenicity. In addition, the prevalence of the virus was monitored by intensified field sampling and PCR analysis from 2016–2020. During this time, the virus was not detected once, thus suggesting a prevalence in the wild of 0.2–0.5%. The low pathogenicity of the local IPNV isolate means that the salmonid populations of fish in Lake Vänern are not immediately threatened.

**Abstract:**

In November 2016, infectious pancreatic necrosis virus (IPNV) was isolated from a broodstock female of landlocked sea trout (*Salmo trutta*) in Lake Vänern in Sweden. VP2 gene sequencing placed the IPNV isolate in genogroup 6, for which pathogenicity is largely unknown. Lake Vänern hosts landlocked sea trout and salmon populations that are endangered, and thus the introduction of new pathogens poses a major threat. In this study we characterized the novel isolate by conducting an infection trial on three salmonid species present in Lake Vänern, whole genome sequencing of the isolate, and prevalence studies in the wild sea trout and salmon in Lake Vänern. During the infection trial, the pathogenicity of the Swedish isolate was compared to that of a pathogenic genogroup 5 isolate. Dead or moribund fish were collected, pooled, and analyzed by cell culture to identify infected individuals. In the trial, the Swedish isolate was detected in fewer sample pools in all three species compared to the genogroup 5 isolate. In addition, the prevalence studies showed a low prevalence (0.2–0.5%) of the virus in the feral salmonids in Lake Vänern. Together the data suggest that the novel Swedish IPNV genogroup 6 isolate is only mildly pathogenic to salmonids.

## 1. Introduction

Infectious pancreatic necrosis virus (IPNV) is an important viral pathogen in farmed Atlantic salmon (*Salmo salar*). IPNV is a non-enveloped, double-stranded RNA virus of the Birnaviridae family with a two-segment genome [1,2]. To date, seven different genogroups have been described, showing a large variability in pathogenicity, with genogroup 5 being the most virulent IPNV in all species of salmonids [3]. Comparative studies of pathogenicity are available in rainbow trout, for example, for genogroup 2, 5, and 6 [4], as well as 1, 5, and 7 [5]. In both studies, the genogroup 5 isolate caused the highest mortalities. Essential to high virulence is the fingerprint Thr, Ala, Thr at positions 217, 221, and 247 of the viral VP2 protein, respectively [6,7,8]. There are no comparative studies available in Atlantic salmon or sea trout. Transmission may occur vertically from the mother, with disease signs emerging in fry around the start of active feeding [9]. IPNV-related mortality can be high in juvenile fish where as many as 10–90% may succumb to infection [10,11,12] depending on fish species and IPNV isolate [2,6], whereas adults are often subclinically infected and thus free from signs of disease [12,13].

Sweden has a widespread network of hydropower dams that acts as migratory barriers, reducing the possibility of natural spawning for diadromous fish. The dams also divide the country into well-defined inland and coastal zones in terms of the presence of fish pathogens. The inland zone is considered free from severe viral infections such as IPNV, viral haemorrhagic septicaemia (VHS), and infectious hematopoietic necrosis (IHN). In the past, Sweden has been spared from IPNV outbreaks in alevin and fry, and only sporadic cases have occurred in farms rearing semi-adult rainbow trout or in returning brood stock salmon or sea trout in restocking farms in the coastal zone. Because of the hydropower dams, natural reproduction of salmonids is not possible in many Swedish rivers, and an extensive re-stocking program is in place to compensate for the lack of natural reproduction. Since IPNV is vertically transmittable, all broodstock females are routinely euthanized after stripping and investigated for the presence of viral infections, as well as the vertically transmittable bacterium *Renibacterium salmoninarum*, which causes bacterial kidney disease (BKD). If a female is infected, the eggs are immediately destroyed.

Lake Vänern is Sweden’s largest lake and has two landlocked populations each of sea trout and Atlantic salmon. The populations have been landlocked since the last ice age. Still, all four populations smoltify at approximately 1–3 years of age and become tolerant to sea salinity levels. They also display anadromous behavior, but migration occurs between the two rivers Klarälven and Gullspångsälven and the lake instead of the sea. Adults generally return for spawning after 2–3 years in the lake. Due to this, in this study we refer to the trout populations as sea trout to acknowledge their origin. Hydropower dams block the migration upstream in both rivers and there is a restocking facility for the four populations. A substantial number of fish are also lifted across the dams in Klarälven each year to allow for recreational fishing and some natural spawning. In Gullspångsälven, there are small areas with suitable spawning habitats and a low level of natural spawning occurs.

In December 2016, one pooled sample from seven Gullspångsälven trout females in the restocking facility tested positive for IPNV by cell culture and ELISA. The result was confirmed by real time and conventional RT-PCR, and sequencing placed the virus in IPNV genogroup 6. The type strain of IPNV genogroup 6 is an isolate from pike (*Esox lucius*) which was demonstrated to be non-pathogenic to rainbow trout and brook trout (*Salvelinus fontinalis*) and have low virulence in pike [14]. Similarly, a Finnish genogroup 6 isolate was shown to have low pathogenicity in rainbow trout [15]. Apart from genogroup 7, genogroup 6 is phylogenetically the most distant group from the other genogroups of IPNV. To determine the pathogenicity of the IPNV isolate in juvenile salmonids, and to evaluate the potential effects of IPNV genogroup 6 on the local salmonid populations, we performed an infection trial. In addition, the prevalence of the virus in the wild populations of salmonids in the Lake Vänern catchment area was monitored during a period of three years. Infection trials in Gullspångsälven sea trout, Atlantic salmon of Klarälven origin, and rainbow trout all indicated that the Swedish genogroup 6 isolate was less pathogenic than the reference isolate Rindsholm (DK-16885-2, genogroup 5). The severity of infection was species dependent and ranged from “mild” in salmon and rainbow trout to “medium” in sea trout. Finally, prevalence studies verified that the Swedish isolate is likely only mildly infective as none of the 900 individuals tested were infected by IPNV. However, there are still potential risks with circulating IPNV infections in the already vulnerable populations of Lake Vänern and the presence of the virus needs to be continuously monitored.

## 2. Materials and Methods

### 2.1. Isolation and Sequencing of the IPNV Isolate

#### 2.1.1. Virus isolation from a Broodstock Farm

At the broodstock facility, spawning females are routinely tested for *Renibacterium salmoninarum* and viral infections post stripping. At the time, sampling and analysis for viruses was performed according to the procedures described in EU Commission Decision 2015/1554. Briefly, once a fish is euthanized, the spleen, kidney, and heart are cut out and one small piece per organ is put in a petri dish. Organ pieces from up to 10 individuals containing, in total, about 1 g of tissue are made. The pool is added to a tube containing approximately 4 mL of Eagles minimal essential medium (EMEM) supplemented with antibiotics and sent chilled to the laboratory. At the laboratory, the samples are transferred to small plastic bags and homogenized using a paddle blender for 30–60 s. Once homogenized, the samples are diluted 1:10 with EMEM, centrifuged for 30 min, and filtered. The cleared lysate is then added onto BF-2 and FHM cells in a 10-fold serial dilution (1:100 to 1:10,000), and the cells are observed every other day for cytopathogenic effects (CPE). All samples, including CPE negative wells, are sub-cultivated on day 7 by transferring 200 μL of conditioned medium to a fresh cell layer, followed by cultivation for an additional 7 days. If no CPE is detected after sub-cultivation, the sample is considered virus negative.

In the case of CPE, the virus is identified at SVA using commercial IPNV, VHSV, and IHV ELISAs (Bio-X Diagnostics, Rochefort, Belgium), according to the manufacturer’s instructions. Results are further verified by real time PCR according to the ELISA results, and nucleotide sequencing is performed to determine the genotype/genogroup. For IPNV, the process is described below.

#### 2.1.2. RNA Extraction and Preparation of cDNA

RNA was extracted from cell cultures using a combination of the TRIzol^®^ Reagent (Invitrogen Thermo Fisher Scientific, Carlsbad, CA, USA) and the RNeasy Mini kit (Qiagen, Hilden, Germany), as described previously [16]. Purified RNA recovered in 40 µL of nuclease-free water and the concentration was measured by the Qubit^®^ fluorometer 2.0 (Invitrogen, Thermo Fisher Scientific, Carlsbad, USA) using the Qubit^®^ RNA Assay Kit (Invitrogen, Thermo Fisher Scientific, Eugene, OR, USA) and stored at −80 °C until further use. Reverse transcription was performed using the SuperScript IV First Strand Synthesis kit (Invotrogen, Thermo Fisher Scientific, Carlsbad, CA, USA) and simultaneous tag-labelling of cDNA using FR26RV-6N random primer was performed. Second-strand cDNA synthesis was performed with 3′→5′ exo-Klenow DNA polymerase (New England BioLabs, Ipswich, MA, USA), followed by 40 cycles of sequence-independent single primer amplification (SISPA) using FR20RV primer [17]. Primer sequences were removed using EcoRV restriction enzyme (New England BioLabs, Ipswich, MA, USA) and a final PCR product clean-up step (QIAquick PCR Purification Kit, Qiagen, Hilden, Germany) was performed, as previously described [16,17,18]. Each sample was amplified in triplicates and pooled to ensure a high enough double-stranded DNA yield after pre-treatment procedures and for the library preparation. After PCR product purification, the concentration of DNA was verified with the Qubit^®^ fluorometer 2.0 and the dsDNA BR Kit (Invitrogen, Invitrogen, Thermo Fisher Scientific, Eugene, OR, USA), and diluted to 0.2 ng/µL according to the Nextera XT DNA library preparation protocol (Illumina, San Diego, CA, USA).

#### 2.1.3. Illumina MiSeq Library Preparation and Sequencing

Prior to Illumina MiSeq sequencing, the amount of virus was increased by inoculation of 0.2 mL of the supernatant onto a fresh layer of cells in a T25 flask. Three Swedish IPNV isolates were sequenced. In addition to the sea trout Vänern isolate from November 2016, two IPNV isolates collected from Atlantic salmon in June 1994 and November 1998, one classified as Serotype Sp (currently genogroup 5) and one that could not be classified according to the serotyping diagnostics present at the time of isolation, were used. DNA was prepared for sequencing using the Nextera XT DNA sample preparation kit (Illumina, San Diego, CA, USA) according to the manufacturer’s instructions with the modifications that half of the recommended amounts of reagent per sample was used, and the library normalization process to equalize the quantity of all libraries was performed by dilution with 10 mM TRIS-HCl (pH 8.5) rather than using beads, as recommended in the Nextera XT protocol. Library quantity and quality was monitored by the Agilent 2100 Bioanalyzer (Agilent Technologies, Waldbronn, Germany). Libraries were normalized by adjusting the molar concentration to 4 nM with 10 mM TRIS-HCl (pH 8.5), pooled, and further prepared for sequencing on the MiSeq platform according to the manufacturer’s instructions (Illumina, San Diego, CA, USA). The pooled library at final a concentration of 12 pM was sequenced as paired-end reads using the MiSeq 600 v3 Reagent Kit (Illumina, San Diego, CA, USA).

#### 2.1.4. Bioinformatics

The paired-end reads were quality trimmed using a Trimmomatic v 0.39 [19] with a sliding window of four nucleotides and required average quality score of 15. The trimmed reads were assembled using SPAdes v 3.15.4 [20]. The segments of IPNV_SWE94_GG2 were assembled with SPAdes in standard operation, while IPNV_SWE98_GG3 and IPNV_SWE16_GG6 were assembled using the “rnaviral” flag. The IPNV contigs were identified using DIAMOND v 2.0.9 [21] with a database for classification, created using the NCBI nr database of GenBank release 248 and the corresponding NCBI taxonomy databases. Phylogenetic analysis followed the approach by Nishizawa et al. [22] and based the analysis on 308 nucleotides encompassing the VP2/NS junction of the polyprotein in segment A of the three newly sequenced isolates together with those of the same set of IPNV sequences used by Nishizawa et al. Sequences (*n* = 91) were aligned, and the alignment was used to create a maximum likelihood (ML) phylogenetic tree using CLC Genomics Workbench v 21 (Qiagen, Hilden, Germany). The ML tree was constructed using a neighbour-joining starting tree and a general time reversible nucleotide substitution model with a transition/transversion ratio of 2. Substitution rates (4) and the gamma distribution parameter were estimated. Bootstrap analysis was performed with 100 replicates.

### 2.2. Experimental Trials

#### 2.2.1. Fish

Experimental infection was performed in accordance with the current animal welfare regulations (Directive 2010/63/EU) and was approved by the Danish Animal Research Authority under the license 2013-15-2934-00976. The rainbow trout fry was reared at DTU facilities from disinfected eyed eggs retailed from an approved IPNV free Danish hatchery. Landlocked sea trout/the Gullspångsälven strain and Atlantic salmon/the Klarälven strain were purchased from the Swedish restocking facility rearing these populations. Low mortality was observed from hatching at the end of April until start of the trial in June. One day prior to infection (9 July 2019), fry was transferred into aerated freshwater 8 L capacity tanks with a constant flow-through system and 300% water renewal per day to acclimatize. Each tank contained 100 fish. Housing conditions were: 12 °C ± 1 °C, light:dark 12:12, an initial stocking density of ≤10 kg/m^3^, feeding 1.5% of biomass. The average weight of the fry at the start of the trial was 0.6 g (rainbow trout), 0.5 g (sea trout), and 0.5 g (Atlantic salmon).

#### 2.2.2. Pre-trial with Rainbow Trout Exposed to Different IPNV Isolates

Four IPNV isolates were tested. All four belonged to genogroup 5. The isolates were St Restrup (DK-16619-3), Christiansminde (DK-15512-2), Hjortlund (DK-17692-1), and Rindsholm (DK-16885-2). A total of 359 rainbow trout fry were used. Fish were split into groups of 59–60 individuals, forming one negative control group and five infection groups. Four infection groups and the negative control group were challenged by intraperitoneal (i.p.) injection of 29–30 fish and cohabitation of 30 fish. The intraperitoneal injection volume was 25 µL (for infection dose see Table 1). Negative control fish were sham injected with cell culture medium (EMEM) with 10% fetal bovine serum (FBS). In all tanks with cohabitants, the injected fish had the adipose fin removed to enable identification. For the DK-16885-2 isolate, infection by immersion was also evaluated. Immersion was performed for 6 h in 4 L water with 2.6 × 10^5^ virus particles. The fry was monitored for 28 days post infection (p.i.), then the pre-trial was ended and six fish from each tank were euthanized for sampling. Details for the pre-trial are summarized in Table 1.

#### 2.2.3. Infection Trial—Regimens and Doses

Based on pre-trial results, IPNV genogroup 5 isolate DK-16885-2 was used, and infection was obtained through an immersion bath for 6 h in 4 L water. The DK-16855-2 isolate was provided by DTU Aqua and had been through 2 passages of cell culture. The Swedish IPNV genogroup 6 isolate, SWE16_GG6, was provided by SVA and had been through 2 passages of cell culture. The infection dose for both isolates was set to 1.3 × 10^8^ virus particles (2.6 × 10^5^ TCID50/mL water). All regimens were run in duplicates, i.e., two tanks of negative controls, DK-16885-2 and SWE16_GG6.

#### 2.2.4. Mortality, Sampling, and Virus Detection

The fish was monitored daily for 33 days p.i. and disease signs and mortality were logged. Moribund fish were euthanized. Moribund and dead fish were collected and sampled for viral culture. In addition, a number of fish were sampled on day 12 p.i. (*n* = 5 per tank) and at the end of the trial (day 34 p.i. (*n* = 10 for negative control, *n* = 5 for DK-16885-2, and *n* = 5 for SWE16_GG6 fish per species)). Morbidity was observed in all tanks containing brown trout two days after starting the trial, with 1–3 dead fish in five of the six tanks (including both duplicate negative control tanks). Morbidity/mortality was not observed in any other tanks at this time. A bacterial infection was suspected, and antibiotic treatment was started immediately in the tanks containing brown trout. Treatment consisted of the addition of 1 mL Norfenicol (Norbrook Laboratories Ltd., Newry, UK) to each tank daily for four consecutive days. Water flow (~2 L/h) was closed for one hour during treatment. Norfenicol contains 300 mg of the active substance (florfenicol) per mL and the tank water volume was ~8 L, thus treatment dose was 37.5 mg florfenikol/L water.

Samples from all moribund or dead fish were tested on EPC and BF-2 cell cultures, as described in the methods section “Isolation of IPNV isolate from a broodstock farm”, and virus was identified by the presence of CPE. Briefly, the heads and tails of the fish were removed (postmortem), and up to 10 individuals, from the same tank and day, were pooled. Tissues were then cut into small pieces and approx. One gram of total tissue was put in cell culture growth media (EMEM), homogenized, and filtered. The filtrate was serially diluted in cell culture growth media before being added to a semi-confluent cell layer of EPC and BF-2 cells, respectively. Cells were inspected under the microscope daily and no CPE was noted. CPE negative cells were sub-cultured for 7 days, as described under “Isolation of IPNV isolate from a broodstock farm”. In the case of CPE, the virus was identified by IPNV ELISA (DTU Aqua SOP Å-4-AR-035). This sandwich ELISA captured the antigen by protein-A purified rabbit anti IPNV (F41 + F51) and detected the virus by Mab Anti IPNV FM-130AY-5 (Amsbio, Abingdon, England) that was visualized by HRP conjugated rabbit anti mouse Ig (Dako P260, Santa Clara, CA, USA).

### 2.3. Prevalence Studies

From November 2016 to November 2020, salmon and sea trout caught by local fishermen and during recreational fishing competitions in Lake Vänern were sampled for prevalence estimations. In addition, a few parr were sampled during electro fishing for population monitoring in the river Klarälven. All samples taken for broodstock health monitoring from the Lake Vänern re-stocking farm (January 2017 to November 2020) were also included. All fish were sampled, and analysis was performed, as described in section “Isolation of IPNV isolate from a broodstock farm” above. Each pool was kept species specific, and at the restocking farm even trout/salmon strain specific. In total, 979 individuals were sampled.

## 3. Results

### 3.1. Virus Isolation during the Initial IPNV Outbreak

Five days after inoculation, one sample collected from Gullspångsälven sea trout on 15 November 2016 showed signs of CPE in two out of the three 10-fold dilutions (1:100 and 1:1000) on the BF-2 cells. At this time, no obvious CPE could be detected on the FHM cells. A few days into sub-culturing, both BF-2 and FHM cells showed CPE. By the end of week two CPE could be observed in all three dilutions for the BF-2 cells, and two out of three dilutions for the FHM cells. Given the nature of the infection, with early CPE in BF-2 and not FHM cells, as well nature of the CPE, IPNV was suspected. The infection was verified by ELISA and real-time RT-PCR.

### 3.2. IPNV Whole Genome Sequencing

Three isolates from cell culture were sequenced. The characterization of IPNV_SWE16_GG6 was the main objective of the present study, while IPNV_SWE94_GG2 and IPNV_SWE98_GG3 provide phylogenetic comparison since there is a scarcity of sequenced Swedish IPNV isolates. Complete coding regions could be obtained for segment A (polyprotein) in all three isolates, while complete coding regions were obtained for segment B (RdRp) of isolates IPNV_SWE16_GG6 and IPNV_SWE98_GG3. A portion at the 5´-end of segment B, representing about fifty amino acids, was lacking for IPNV_SWE94_GG2. The sequences have been deposited in GenBank with accession numbers ON645941 and ON645942 for IPNV_SWE94_GG2, ON624133 and ON624134 for IPNV_SWE98_GG3, and finally ON409687 and ON409688 for IPNV_SWE16_GG6. Assignment to genogroups was performed by phylogenetic comparison of the sequence at the VP2/NS junction of the polyprotein [22]. IPNV_SWE94_GG2, IPNV_SWE98_GG3, and IPNV_SWE16_GG6 were assigned to genogroup two, three, and six, respectively (Figure 1). The phylogenetic topology and genogroup assignment for the previously sequenced isolates were, in all cases, in accordance with previous results [22].

### 3.3. Experimental Infection by IPNV

#### 3.3.1. Sea Trout

As the genogroup 5 isolated DK-16855-2 showed the highest mortality in the pre-trial, this isolate was used as a high pathogenicity control for the infection trial in all species. A total of 454 fish entered the trial divided into 6 tanks separated as follows: 152 control fish, 164 exposed to the DK-16885-2 isolate, and 137 exposed to the SWE16_GG6 isolate. Sea trout mortality started on day two in all tanks including the control tanks, indicating infection with a non-IPNV pathogen. A bacterial infection was suspected, and antibiotic treatment was started. In total, 169 pools were tested, but 4 of them were excluded from analysis as they contained more than 10 individuals. Through the course of the trial, 290 individuals died, 77 of them in control tanks (Table 2).

One of the 61 tested pools, originating from control tanks and collected on day 19 p.i., was positive for IPNV. This pool was collected simultaneous with a pool of sea trout infected with DK-16885-2 which was negative, and it is beyond doubt that the unexpected outcome was due to a mismatch between the two samples. No other control tank samples were positive. Sequencing revealed the virus as IPNV genogroup 5 (i.e., the DK-16885-2 isolate). For the DK-16885-2 isolate, IPNV-induced mortality started on day six and increased in a somewhat biphasic pattern throughout the trial (Figure 2A). In total, 32 out of 55 pools were positive for DK-16885-2, all but two without any sub-cultivation, indicating a high virus load in the samples (Figure 2B). For the SWE16_GG6 isolate with unknown pathogenicity, 11 out of 49 tested pools were positive for IPNV, four of which were positive during primary cultivation (Figure 2 A,B).

Total number represents the number of fish in the tank at the time of viral challenge (day 0). Survivors represent the number of fish in the tank when the trial was ended (day 34 post infection).

#### 3.3.2. Rainbow Trout

A total of 583 fish entered the trial separated as follows: 189 control fish, 197 exposed to the DK-16885-2 isolate, and 197 exposed to the SWE16_GG6 isolate. No significant mortalities were observed in the control tanks (Table 3).

In general, the rainbow trout was resistant to severe IPNV infection and only 19 fish died. Therefore, not only moribund fish was sampled, but also individuals that were deviating in behaviour or coloration. IPNV positive samples started to be identified around day ten, but quickly reached a plateau (Figure 3A). Out of the 66 pools tested, 29 were positive for IPNV and 25 out of them were from the DK-16885-2 infection group. Only three pools were IPNV positive after infection with the SWE16_GG6 isolate. All IPNV positive pools induced CPE during primary cultivation (Figure 3B). All in all, one fish died in the negative control groups on day 7 p.i. and the sample was identified as IPNV positive. Sequencing revealed that this sample also contained the DK-16885-2 isolate. Similarly, this was due to a cross contamination with DK-16885-2 during handling of the single fish or sample. No other control group fish were IPNV positive.

Total number represents the number of fish in the tank at the time of viral challenge (day 0). Survivors represent the number of fish in the tank when the trial was ended (day 34 post infection).

#### 3.3.3. Atlantic Salmon

A total of 495 fish entered the trial separated as follows: 175 control fish, 169 exposed to the DK-16885-2 isolate, and 151 exposed to the SWE16_GG6 isolate. One infected tank of the SWE16_GG6 isolate and control tank experienced high mortality rates starting on day 22 and 20, respectively, of the trial, but for Atlantic salmon no antibiotics were administered. By the end of the trial, a total of 145 fish (57 in control tanks, 19 in DK-16885-2 isolate tanks, and 69 in SWE16_GG6 isolate tanks) had died (Table 4). However, virus analysis showed that few of the mortalities were due to IPNV infection as only three DK-16885-2 group pools and one SWE16_GG6 group sample were IPNV positive (Figure 4).

Total number represents the number of fish in the tank at the time of viral challenge (day 0). Survivors represent the number of fish in the tank when the trial was ended (day 34 post infection).

### 3.4. IPNV Prevalence in the Wild

During November 2016–November 2020, sea trout and Atlantic salmon were caught and tested for the presence of IPNV both in Lake Vänern and upstream in Klarälven. It total, 124 pools, of which 117 pools contained in total 900 individuals (341 sea trout, 552 Atlantic salmon, and 7 individuals that were either sea trout or Atlantic salmon) and 7 pools contained an unknown number of individuals (*n* = 1–10), were tested by cell culture (Table 5). No pool tested positive for IPNV. Including the 79 sea trout broodstock samples from 2016, where one pool including 7 fish which was positive, produces a sample size of 979 individuals plus 7–70 individuals from the “unknown” pools. We assume that the positive pool only included one infected female because it is unlikely that >1 infected individual ends up in the same pool with so many negative samples collected. Thus, the prevalence in our sampled population is approximately 0.1%. In a population of >25,000, the sample size required to find at least one positive individual (with 95% confidence) at 0.1% prevalence is 2822 [23]. The estimated total prevalence is 0.2–0.5%, as our sample size is between the sample sizes needed for those prevalences (*n* = 1452 and *n* = 591, respectively) [23]. Sampling of broodstock has continued after this study, and none have been infected with IPNV, but the next level in sample size has not yet been reached.

## 4. Discussion

In this study we investigated the pathogenicity of a novel IPNV isolate in three species of salmonids: landlocked sea trout, landlocked Atlantic salmon, and rainbow trout. Sequencing placed the isolate in genogroup 6, a genogroup never identified in Sweden and whose pathogenicity that had not previously been characterized in sea trout or Atlantic salmon. Our data presented herein show that the novel Swedish genogroup 6 isolate, SWE16_GG6, should be considered as a low pathogenicity type of IPNV, compared to the known highly pathogenic “DK-16885-2” of genogroup 5 and serotype Sp. [15].

The total landlocked salmon and sea trout population in the lake is small (approximately 2500 individuals return for spawning annually) and genetically unique and as such is vulnerable to pathogens and other impacts on their habitat. Because several hydropower dams prevent natural migration and spawning, an extensive restocking program is in place, in addition to manual lifting and transportation of individuals upstream of the several hydropower dams. If a highly pathogenic and vertically transmitted disease is introduced to the populations, the assisted migration could pose a significant risk for the population’s survival as it would move the pathogen upstream to potentially have serious effects on the natural regrowth of the populations. In addition, infection in broodstock leads to the destruction of potentially infected eggs, and thereby there are fewer fish for restocking. Thus, the impact of a highly pathogenic IPNV isolate for the survival of the combined salmon and sea trout population in Lake Vänern could rapidly become devastating. Further, Lake Vänern houses a net pen rainbow trout farm. Introduction of a virus to that farm would mean an increased infection pressure within the lake, and if fish is sold to other farms or for put and take before the disease is detected, it would allow the spread of the virus into new water systems with other salmonid populations. As a precaution, all potentially infected eggs were immediately destroyed already in 2016 and no investigation of the virulence of the Swedish isolate on the offspring was thus possible at the time. Instead, the pathogenicity was investigated later in newly hatched fry of the Gullspångsälven population of landlocked seatrout and the Klarälven population of landlocked Atlantic salmon. The aim of this study was to (1) evaluate the potential effects of this IPNV genogroup on the local, endangered sea trout and salmon populations of Lake Vänern, (2) evaluate the potential effects of this IPNV isolate on farmed rainbow trout, and (3) to determine the prevalence of IPNV in the wild populations of salmon and sea trout in Lake Vänern.

To investigate the pathogenicity of the SWE16_GG6 isolate, we used cell culture of pooled samples for the detection of virus instead of qPCR. Cell culture detects infectious virus particles, whereas qPCR detects viral nucleic acid regardless of whether there is infection or just the presence of virus DNA/RNA. Unfortunately, pooling of samples makes it impossible to say how many of the fish are infected, and thus makes it difficult to compare how much more pathogenic an isolate is compared to another. Still, this methodology is commonly used in infection trials because a laboratory setting is far from natural, and it is often sufficient to classify new isolates as “more or less pathogenic” than the currently known strains. The study comprised of 1532 fish of less than 1 g, but examining all of these individuals was not possible in the given set up. Here, SWE16_GG6 was compared to a known highly pathogenic isolate of genogroup 5, DK-16885-2. No strain with low pathogenicity, such as a genogroup 2 strain, was included as the aim was to decide whether the new isolate was a highly pathogenic isolate or not. The data clearly showed that the DK-16885-2 isolate caused higher infection prevalence in both sea and rainbow trout than the SWE16_GG6 isolate. Both isolates had limited pathogenicity in salmon. In fact, only one pool of salmon samples was positive for the Swedish isolate. This indicates a high resistance to the virus and the natural host for the SWE16_GG6 isolate is likely one of the landlocked salmon populations. In addition, fish were found to be positive for the DK-16885-2 isolate throughout the trial once infection was established in a tank, whereas the SWE16_GG6 isolate was only detected at a few sampling points. This indicates either a lower level of virus replication in infected individuals or a shorter duration of infection for the SWE16_GG6 isolate, both indicative of a lower pathogenicity. Unfortunately, due to a likely bacterial infection in the tanks with sea trout, and to some extent Atlantic salmon, we cannot compare mortality rates between the different species. However, only a few rainbow trout died during the trial despite a high number of IPNV (DK-16885-2) positive pools. This is indeed very surprising as rainbow trout normally is highly sensitive to IPNV, and in the pre-trial none of the rainbow trout in the DK-16885-2 immersion group survived more than 20 days post infection. For the SWE16_GG6 isolate, only three pools originating from rainbow trout were positive. The two findings of IPNV in samples collected from control tanks with sea trout and rainbow trout, respectively, were both attributed to a mis-match between two samples or cross contamination. All other samples from negative control tanks were negative.

A recent study compared the pathogenicity of several Finnish IPNV isolates in rainbow trout, including one genogroup 6 isolate [4]. Like our results, they observed a low mortality rate in rainbow trout, and thus a low pathogenicity of IPNV. This is not very surprising given the high similarity of the SWE16_GG6 and Finnish genogroup 6 (KY548509) isolates with an overall amino acid homology of 94.65%. In the VP2 protein of 422 amino acids, a protein involved in the initial infection by the virus, the similarity is even higher (99.3%). Both isolates have Pro at position 217 and Ala at position 221, which have been shown to give an “avirulent” IPNV isolate in salmon [7,24]. The Finish isolate is also the closest related IPNV isolate of genogroup 6 with an 83% nucleotide identity of VP2. Surprisingly, when sequencing the DK-16885-2 isolate obtained from the “IPNV positive negative tanks”, tank 21 and 15, we also noticed the same amino acid composition at position 217 and 221 (Pro and Ala, respectively). This likely explains why we noticed a discrepancy to the pre-trail, where the DK-16885-2 isolate caused a 100% mortality. Unfortunately, we do not know if this mutation occurred during the trial or if the batch of virus used in the trial already contained the mutations.

In this study, we do not only present data for a newly discovered IPNV isolate, but also for the first time present data on the pathogenicity of an of IPNV virus of genogroup 6 in several different species of salmonids. Infection trials suggest that genogroup 6 should be considered as a low pathogenic type of IPNV for all three salmonid species tested. In addition, our extended prevalence study that included 979 individuals sampled over five years, where all except the originally infected fish tested negative for IPNV, further indicates a low pathogenicity of the isolate. In fact, to date the virus has not been isolated again since its discovery in 2016. Therefore, the risk of spreading IPNV by selling or moving farmed rainbow trout from the Lake Vänern catchment area is minimal. Furthermore, the risks, or likelihood, of the virus getting a foothold in the populations of landlocked sea trout and Atlantic salmon should also be considered as small. This study was designed to investigate pathogenicity when all fish were simultaneously exposed to the virus and did not answer how well the virus spreads within the population after initial infection, especially among adults. Also, it cannot estimate how likely this variant of IPNV would transfer vertically. These are, therefore, unknown factors, and potential risks, when dealing with the virus in Lake Vänern. In addition, individuals that are lifted past the dams could be infected, thus spreading the virus upstream and, over time, reducing natural reproduction. As populations undergo assisted reproduction, we do hope that if the virus manages to gain a foothold in the population, we will notice it in time in the broodstock. Thus, continuous sampling is necessary to conserve endangered populations

## 5. Conclusions

The combined data in this study show that the IPNV genogroup 6 isolated in 2016 from a sea trout female of Lake Vänern should be considered as a low pathogenicity isolate in salmon and rainbow trout, but medium in sea trout. 

## Figures and Tables

**Figure 1 vetsci-10-00058-f001:**
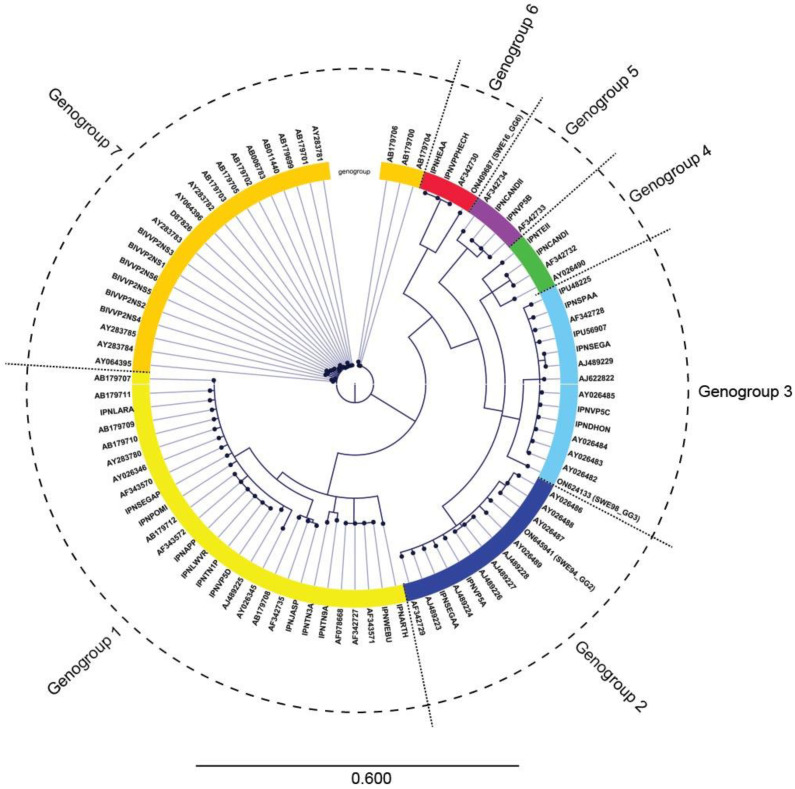
Circular maximum likelihood phylogram of 308 nucleotides of the VP2/NS junction of the polyprotein segment of ninety-one IPNV isolates. All sequences have been named with their GenBank accession, except the three Swedish isolates that in addition have been denoted by the isolate names within parentheses. Genogroup 1 (yellow); genogroup 2 (dark blue); genogroup 3 (light blue); genogroup 4 (green); genogroup 5 (purple); genogroup 6 (red), and genogroup 7 (orange).

**Figure 2 vetsci-10-00058-f002:**
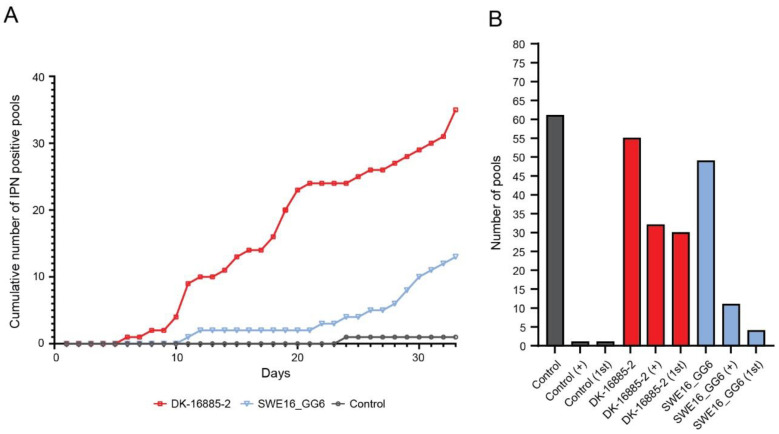
Investigation of the pathogenicity of IPNV in sea trout. (**A**) the cumulative number of IPNV positive pools by cell culture for the DK-16885-2 (red) and the SWE16_GG6- (blue) isolates. (**B**) the total number of pools tested, and the number of pools positive for IPNV. Pools positive for IPNV before subculturing are indicated by “1st”, and total number of positive pools are indicated by “+”. Control samples are showed in grey; DK-16885-2 isolate in red, and Swedish isolate in blue.

**Figure 3 vetsci-10-00058-f003:**
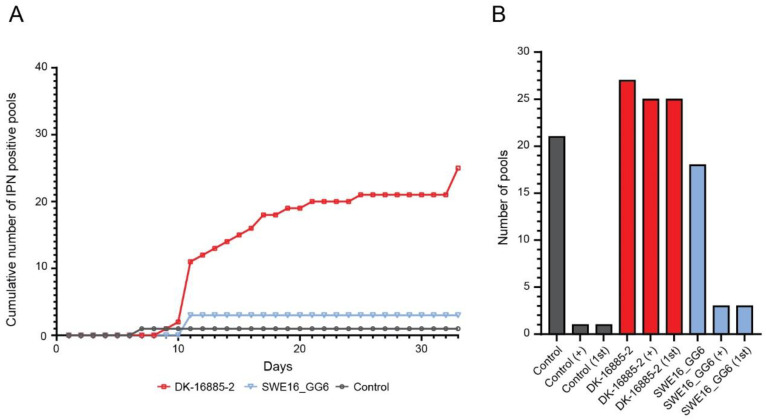
Investigation of the pathogenicity of IPNV in rainbow trout. (**A**) the cumulative number of IPNV positive pools by cell culture for the DK-16885-2 (red) and the SWE16_GG6- (blue) isolate. (**B**) the total number of pools tested and the number of pools positive for IPNV. Pools positive for IPNV before subculturing are indicated by “1st”, and total number of positive pools are indicated by “+”. Control samples are showed in grey; DK-16885-2 isolate in red, and Swedish isolate in blue.

**Figure 4 vetsci-10-00058-f004:**
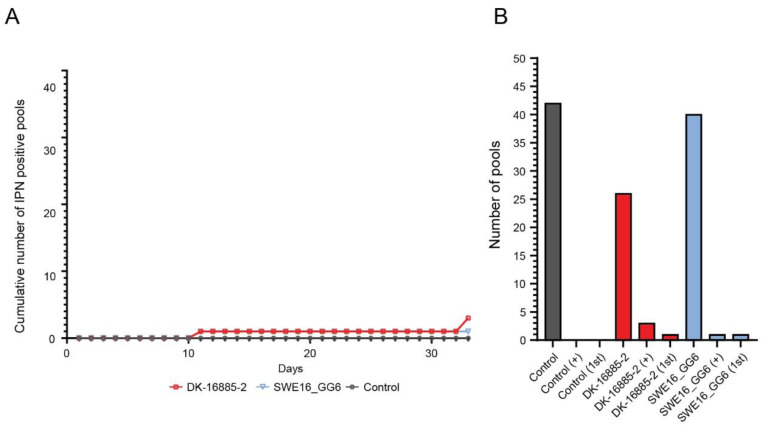
Investigation of the pathogenicity of IPNV in Atlantic salmon. (**A**) the cumulative number of IPNV positive pools by cell culture for the DK-16885-2- (red) and the SWE16_GG6- (blue) isolate. (**B**) the total number of pools tested and the number of pools positive for IPNV before or after subculturing. Pools positive for IPNV before subculturing are indicated by “1st”, and total number of positive pools are indicated by “+”. Control samples are showed in grey; DK-16885-2 isolate in red, and Swedish isolate in blue.

**Table 1 vetsci-10-00058-t001:** Details for IPNV pre-trial.

Group	Virus Strain	Infection Route	Number of fish			Infection Dose (TCID50/mL) *	Infection Dose/Fish
			Injected	Cohabitants	Immersion		
NC	Negative control	i.p + cohab	29	30	-	0	0
StR	16619-3	i.p + cohab	30	30	-	5.9 × 10^4^	1.48 × 10^3^
C	15512-2	i.p + cohab	30	30	-	8.6 × 10^5^	2.15 × 10^4^
H	17692-1	i.p + cohab	30	30	-	1.9 × 10^5^	4.75 × 10^3^
R-inj	16885-2	i.p + cohab	30	30	-	1.3 × 10^8^	3.25 × 10^5^
R-imm	16885-2	immersion	-	-	60	2.6 × 10^5^	

* The TCID50/mL infection dose is per mL injected fluid for the injected groups and per ml water for the immersion group.

**Table 2 vetsci-10-00058-t002:** Infection trial data on sea trout.

	Tank 4	Tank 21	Tank 5	Tank 20	Tank 6	Tank 19	Total
Treatment	Control	Control	DK-16885-2	DK-16885-2	SWE16_GG6	SWE16_GG6	
Total number	80	73	75	89	49	88	454
Survivors	56	20	41	2	20	25	164
Moribund	24	53	34	87	29	63	290

**Table 3 vetsci-10-00058-t003:** Infection trial data on rainbow trout.

	Tank 10	Tank 15	Tank 11	Tank 14	Tank 12	Tank 13	Total
Treatment	Control	Control	DK-16885-2	DK-16885-2	SWE16_GG6	SWE16_GG6	
Total number	85	104	98	99	100	97	583
Survivors	84	103	93	90	98	96	564
Moribund	1	1	5	9	2	1	19

**Table 4 vetsci-10-00058-t004:** Infection trial data on Atlantic salmon.

	Tank 7	Tank 18	Tank 8	Tank 17	Tank 9	Tank 16	Total
Treatment	Control	Control	DK-16885-2	DK-16885-2	SWE16_GG6	SWE16_GG6	
Total number	82	93	94	75	84	67	495
Survivors	75	43	83	67	29	53	350
Moribund	7	50	11	8	55	14	145

**Table 5 vetsci-10-00058-t005:** Prevalence study regarding IPNV in the salmon and sea trout populations inhabiting Lake Vänern.

Species	Population	Upstream ^1^	Lake Vänern ^2^	Total
Atlantic salmon	Klarälven	131	98	229
Atlantic salmon	Gullspång	153	46	199
Atlantic salmon	Unknown	0	124	124
Sea trout	Klarälven	155	26	181
Sea trout	Gullspång	152	14	166
Sea trout	Unknown	0	73	73
Mixed	Unknown	0	7	7
Total		591	388	979

^1^ Upstream represent all samples taken at the broodstock facility from the first identification of IPNV in Nov 2016 to the end of the breeding season in 2020. ^2^ Samples collected by fishermen or during recreational fishing competitions from June 2017 to April 2019.

## Data Availability

The sequences have been deposited in GenBank with accession numbers ON645941 and ON645942 for IPNV_SWE94_GG2, ON624133 and ON624134 for IPNV_SWE98_GG3, and finally ON409687 and ON409688 for IPNV_SWE16_GG6. Additional data is available upon request.

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
