# Peer review of "Characterization of a Novel Infectious Pancreatic Necrosis Virus (IPNV) from Genogroup 6 Identified in Sea Trout (Salmo trutta) from Lake Vänern, Sweden"

_vetsci, 2023, doi:10.3390/vetsci10010058_

Round 1

Reviewer 1 Report

This study isolated an IPNV of genogroup 6 from sea trout in Sweden. The isolated IPNV isolate is relatively low pathogenic to three species tested compared to a genogroup 5 IPNV. The results are interesting. There are several issues need to be addressed.

1. has the genogroup IPNV been reported before in Sweden.

2. can the authors presented the sequence analysis of this isolate with other genogroup 6 isolates. How much amino acids are different.

3. why the number of fish in each pool is different.

4. are there any characterization observed in tissues of the fish upon the genogroup 5 and genogroup 6 IPNV infection.

Author Response

  1. has the genogroup IPNV been reported before in Sweden.

-No, the genogroup has never been identified in Sweden. We have clarified that in the text.

  1. can the authors presented the sequence analysis of this isolate with other genogroup 6 isolates. How much amino acids are different.

-The amino acid identity between the DK-16885-2 (genogroup 5) and SWE16-GG6 (genogroup 6) is 86.6% identical, 95.4% similar using L-align of the 388 amino acids known for DK-16885-2. We have added this to the discussion. The VP2 segment that was first analyzed showed maximum identity to a Finnish Genogroup 6 isolate (83%, KY548508)

  1. why the number of fish in each pool is different.

-The fish tanks were inspected twice daily, and dead or moribund fish collected/pooled immediately. Therefore, the pool sizes differ.

  1. are there any characterization observed in tissues of the fish upon the genogroup 5 and genogroup 6 IPNV infection.

-the size of the fish in the trial were too small (~0.5g) to make any observations on the tissue.

Reviewer 2 Report

In this manuscript, the authors have identified a novel infectious pancreatic necrosis virus 1 (IPNV) genogroup 6 in sea trout and compared its pathogenicity in wild sea trout, rainbow trout and salmon in lake Vanern. The data suggest that the novel Swedish IPNV genogroup 6 isolate is only mildly pathogenic to salmonids. However, there are some points are not clear enough and are listed below:

1、  The survivors in different tanks vary greatly (Table 2 and Table 4). The repeatability is not very good. So, how did the authors confirm the pathogenicity of the virus?

2、 In Table 5what does mixed stand for, salmon and trout? And the number was just 7.

3、  Line 286, “four of which were positive during primary cultivation (Figure 1A-B).” Did Figure I have A and B?

4、  Why was DK-16885-2 selected for infection trial?

5、  Why was the DK-16885-2 not included in the phylogenetic tree (Figure 1)?

6、  What are the differences between different genogroups? How about the pathogenicity? Is there any relationship between pathogenicity and fish species? The background information should be included in the introduction section.

Author Response

  1. The survivors in different tanks vary greatly (Table 2 and Table 4). The repeatability is not very good. So, how did the authors confirm thepathogenicity of the virus?

-Unfortunately, we a bacterial infection occurred in some of the tanks and antibiotics was needed. Therefore, we could not compare the number of dead fish after IPNV exposure. Instead, all the fish that died during the trial was pooled and analyzed by cell culture, also in “uninfected” tanks. Pathogenicity in this trial is referred to as fish dying with IPNV, but strictly speaking the cause of death might not have been IPNV. However, IPNV was likely a contributing factor. This is in a way like the definition used for COVID-19 statistics during the pandemic.

  1. In Table 5,what does mixed stand for, salmon and trout? And the number was just 7.

-Yes, it means that it is a pool consisting of organs from booth salmon and trout. It is unknown how many of each.

  1. Line 286, “four of which were positive during primary cultivation (Figure 1A-B).” Did Figure I have A and B?

-This is unfortunately a mistake from rearranging the figures. That should refer to figure 2A-B.

  1. Why was DK-16885-2 selected for infection trial?

-This is mentioned on row 191 in the submitted manuscript. “Based on pre-trial results, IPNV genogroup 5 isolate DK-16885-2 was used, and infection made through immersion bath 6 h in 4 L water“. We have emphasized this further and added this information also to the results section. This has been added to the manuscript. DK-16885-2 was used based on the data from the pre-trial where DK-16885-2 had the highest pathogenicity.

  1. Why was the DK-16885-2 not included in the phylogenetic tree (Figure 1)?

-The sequence analysis for the Genogroup 6 isolate was performed before the start of the trial when the sequence of DK-16885-2 was not available. VP2 of DK-16885-2 was sequenced first once IPNV positive pools occurred in negative tanks the trial. We decided not to include the sequence in the analysis as the sequence is incomplete (first 388 amino acids). After discussions we decided not to add that sequence to figure 1as that it will not add anything to the outcome of the study. If needed for publication, we can add the partial DK-16885-2 sequence to the analysis.

  1. What are the differences between different genogroups? How about the pathogenicity? Is there any relationship between pathogenicity and fish species? The background information should be included in the introduction section.

-Historically genogroup 5 has shown the highest pathogenicity. Unfortunately, many of the trials that has been conducted are in rainbow trout. For Genogroup 6, only one trial has been published and we are in the discussion comparing our data to that trial. This finish trial is in rainbow trout, and they noticed a similar pattern to us, thus verifying IPNV genogroup 6 as a “mild” type of disease. We have added a section to the introduction describing what is known about the pathogenicity for IPNV in different species of salmonids.
